# *Brassica tardarae* (Brassicaceae), a New Species from a Noteworthy Biotope of South-Western Sicily (Italy)

**DOI:** 10.3390/plants9080947

**Published:** 2020-07-27

**Authors:** Vincenzo Ilardi, Angelo Troia, Anna Geraci

**Affiliations:** 1Dipartimento di Scienze della Terra e del Mare (DiSTeM), Università degli Studi di Palermo, 90123 Palermo, Italy; vincenzo.ilardi@unipa.it; 2Dipartimento di Scienze e Tecnologie Biologiche, Chimiche e Farmaceutiche (STEBICEF), Università degli Studi di Palermo, 90123 Palermo, Italy; anna.geraci@unipa.it

**Keywords:** Mediterranean flora, morphological variability, endemism, Tardara Gorges, speciation center, Cruciferae

## Abstract

A new species of *Brassica* sect. *Brassica* is described here from Sicily (Italy), which is known to be one of the centers of the diversification of wild taxa of this group. The new species (named *Brassica tardarae*) is restricted to the carbonate cliffs in the Tardara Gorges between Menfi and Sambuca di Sicilia (Agrigento province), an area with a peculiar geological history and where another strictly endemic species was recently described. The morphological relationships between the new species and other similar taxa are discussed, and an analytical key to the Sicilian taxa belonging to the genus *Brassica* sect. *Brassica* is also provided.

## 1. Introduction

*Brassica* is one of 321 genera of the Brassicaceae family, Tribe Brassiceae [1], comprising very variable species, both morphologically and genetically, which are widely utilized for human nutrition, mainly as vegetables, condiments, and edible oils. The genus *Brassica*, including 38 annual or perennial species [1], is divided into three sections: (1) sect. *Brassica*, (2) sect. *Brassicaria*, and (3) sect. *Melanosinapis* [2].

*Brassica* sect. *Brassica* consists of 14 species (with several subspecies) characterized by x = 9 genome, a suffrutescent perennial habit, large size (up to two meters tall when flowering), flowers with yellow–white petals, cylindrical or subcylindrical/tetragonous fruits (siliques), globose and reticulate seeds. The life cycle is usually between two and five to six years [3]. The populations occur mainly around the Mediterranean basin, from Israel and Lebanon in the East to the Canary Islands and the British Isles in the West [4]. On the basis of the literature, the species of this group are *B. oleracea* L., *B. montana* Pourr., *B. bourgeaui* (Webb) Kuntze, *B. cretica* Lam., *B. hilarionis* Post, *B. incana* Ten., *B. botterii* Vis., *B. macrocarpa* Guss., *B. rupestris* Raf., *B. villosa* Biv., *B. insularis* Moris, *B. atlantica* (Coss.) O. E. Schulz, *B. tyrrhena* Giotta, Piccitto & Arrigoni, and the recently described *B. trichocarpa* C. Brullo, Brullo, Giusso& Ilardi.

The morphological characteristics of these taxa show great variability within the populations. Some taxa were indeed split into new species or subspecies, with some populations described as new taxa, mostly in the last forty years [4,5,6,7,8,9,10,11]. At the same time, the ability of these taxa to intercross and generate greater or fewer fertile hybrids led to them being considered as few natural taxa, even up to the extreme of recognizing only one species [12,13].

As is currently known, Sicily is one of the centers of the diversification of wild taxa of *Brassica* sect. *Brassica* [3,4]. The 12 taxa of the section occurring in Sicily and in the small islands around are *B. incana* (in eastern Sicily), *B. insularis* (Pantelleria), *B. macrocarpa* (endemic to the Egadi Islands), *B. rupestris* (with two subspecies: subsp. *rupestris* and subsp. *hispida* Raimondo et Mazzola), and *B. villosa* (with five subspecies: subsp. *villosa*, subsp. *bivonana* (Mazzola et Raimondo) Raimondo et Mazzola, subsp. *drepanensis* (Caruel) Raimondo et Mazzola, subsp. *tineoi* (Lojac.) Raimondo et Mazzola, subsp. *brevisiliqua* (Raimondo et Mazzola) Raimondo and Geraci). More recently, new species were described on the basis of their morphological traits, namely, *B. raimondoi*, which occurs only in the cliffs of Castelmola (Messina) [10] and was recently considered a subspecies of the sympatric *B. incana* [14], if not synonymous, and *B. trichocarpa*, which is endemic to a single locality near Mt. Cuccio, close to Palermo [11].

*Brassica rupestris* and *B. villosa* are endemic to western Sicily [4,7,9,15], with a single disjunct population of *B. rupestris* in Calabria [15]. All populations of these taxa grow on cliffs and rocky habitats (mainly on limestone) from the sea level to 1000-1200 m above sea level (a.s.l.). Recently, Tatout et al. [16] proposed considering *B. villosa* subp. *drepanensis* as a distinct species on the basis of molecular analyses, hypothesizing a hybrid origin of this taxon involving *B. incana* and *B. villosa*.

Since most of the taxa are endemic and/or with a restricted or highly fragmented area of distribution, their populations are often threatened by anthropic activities, grazing, fires, etc. *Brassica macrocarpa* is considered to be Critically Endangered (CR) [17,18,19], whereas *B. rupestris* s.l. and *B. villosa* s.l. are included as Near Threatened (NT) species on the IUCN Red List [20,21]. In previous national and regional Red Lists, according to the IUCN categories, these species were assessed as Lower Risk (LR) (*B. rupestris*. subsp. *rupestris*, *B. villosa* subsp. *bivonana*, and *B. villosa* subsp. *drepanensis*), Vulnerable (VU) (*B. villosa* subsp. *tineoi* and *B. villosa* subsp. *brevisiliqua*), Endangered (EN) (*B. rupestris* subsp. *hispida*), and Critically Endangered (CR) (*B. villosa* subsp. *villosa*) [22,23].

These wild taxa, belonging to the same cytodeme of *Brassica oleracea*, comprising crop species such as cabbage, cauliflower, broccoli, Brussels sprouts, and kale, can hybridize with cultivated forms to represent a useful genetic resource for the improvement of cultivated varieties, considering the importance of *Brassica* vegetables globally [4,12,24,25,26,27,28,29].

During field investigations in south-western Sicily, the first author noticed and for years observed a population of *Brassica* located at “Gole della Tardara”, a Gorge about 11 km North–NorthWest from the town of Sciacca (Agrigento province). Further investigations by the current authors led to the conclusion that this population was different from all other known similar taxa, therefore, it is described here as a new species.

## 2. Results

### 2.1. Morphological Analysis

Table 1 describes the main features of the Tardara Gorges *Brassica* population and compares them with those of the most similar taxa, i.e., *B. rupestris* (subsp. *rupestris* and subsp. *hispida*) and *B. villosa* subsp. *brevisiliqua*. The studied population seems to be very close to these taxa regarding the indumentum of the leaves, which are glabrous, or with rare bulbose hairs, but it differs from *B. rupestris* in terms of the very short fruit (up to half the length) with an evident dorsal rib. The fruiting pedicels are long, making the ratio of the siliqua to the pedicel rather small compared to the other taxa. The petals are smaller than in *B. rupestris* and similar to the ones of *B. villosa* subsp. *brevisiliqua* for both size and color. Nevertheless, the studied population differs from *B. villosa* subsp. *brevisiliqua* in terms of the morphology and thickness of the basal leaves, the diameter of the seeds, and the habit of the reproductive plants, since the former shows long flowering and fruiting branches.

Concerning the micromorphological observations on the leaf, pollen, and seed surfaces, the studied population shows bulbose hairs and numerous stomata in the lower blade (694/mm^2^). Pollen grains are elliptical, tapered at the poles, and bigger than in the other compared taxa, with a size similar to those of the *B. villosa* group [30].

### 2.2. Taxonomic Outcome

According to our observations on morphology and phenology, and our comparative analysis, the Tardara Gorges *Brassica* population must be referred to a new species, hence the description herein.

#### *Brassica tardarae* Ilardi, Geraci and Troia, sp. nov.

Diagnosis: Planta *Brassicae rupestri* similis, sed siliqua crassiore et fere dimidio breviore (sicut in *Brassica villosa* s.l.), rostro breviore, pedicello fructifero longiore, petalis minoribus et luteolis, polline maiore (sicut in *Brassica villosa* s.l.), stomatibus duplis in foliae inferiore pagina, differt.

Typus: Gole della Tardara, territorio di Sambuca di Sicilia (provincia di Agrigento), *in rupibus calcareis*, Ilardi, Troia, Geraci, 9 May 2019 (Holotypus PAL 109799).

Description (Figure 1): suffrutex, 70−150 (180) cm tall, with a robust woody green stem, branching from the base with a diameter of 2–2.5(–3) cm. Leaves glabrous, sometimes (in the young or sterile shoots) with sparse bulbose hairs mainly in the upper blade. Basal leaves petiole 7–14 cm, lamina length 15–23 × 9–14 cm width, ovate–elliptic, sublyrate. Petiole with 1−2 pairs of small lobes in the upper (distal) part; the apical leaf lobe acute with margin irregularly (more or less deeply) toothed. Cauline leaves: gradually smaller, undivided, lanceolate, shortly petiolate to sessile in the upper stem, the biggest with a lamina of 5.5–7 × 1.2–1.5 cm. Inflorescence up to 100–120 cm long, paniculate, pyramidate, with long branches from the base. Sepals 7.0–7.5 mm long × 1.9–2.2 mm wide, petals spatulate 13–18 × 6.5–8 mm, light yellow, claw 3–7 mm long. Siliquae patent to erect–patent, tetragonous, with thick valves, slightly compressed (21)–26–36 (41) × 4.8–6.5 mm, beak 4–5.5 mm long, cylindrical, pedicels 16–26 mm long. Seeds subglobose, diameter 2.25–2.6 mm, reticulate, brown–blackish with small light streaks at the hilum, arranged in a single row in each loculus.

Phenology: Flowering late January to early February, fruiting mid-May.

Distribution and ecology: The species is limited to the limestone cliffs of the “Gole della Tardara” in Sambuca di Sicilia territory (Agrigento, Sicily, Italy), between 100 and 400 m a.s.l. and about 10 km from the sea coast (Figure 2). It lives in a chasmophytic community, described as the endemic association between *Brassico rupestris*-*Centauretum saccensis*, where it is one of the characteristic species [31]. The geographical, climatic, bioclimatic, and vegetational descriptions of this biotope can be found in Bazan et al. [31] and Raimondo et al. [32].

Conservation status: Although its habitat is naturally protected from most human activities (but not from quarries, for example), the unique population (with an area of occupancy of about 8 km^2^, and a single location) is “prone to the effects of human activities or stochastic events within a very short time period in an uncertain future, and is thus capable of becoming Critically Endangered or even Extinct in a very short time period” [33]. On the basis of currently available data, we apply the “D” criterion [33], assessing the species as Vulnerable (VU D2).

Eponymy: The specific epithet refers to the name of the gorges (see above) where the taxon lives.

### 2.3. Analytical Key to the Taxa of Brassica sect. Brassica in Sicily

This key is largely modified from those proposed by previous authors [7,10,15].
1. Leaves glabrous or hispid with bulbose, scattered hairs; 21. Leaves villous or pubescent;72. Petals white; siliqua laterally compressed; *B. insularis*2. Petals yellow; siliqua isodiametric or dorsally compressed;33. Leaf teeth obtuse; siliqua navicular isodiametric, 25–40 mm × 8–12 mm; valves thickened, spongy, dorsally smooth; rostrum widely conic, 10–15 mm, 1–2-seeded;*B. macrocarpa*3. Leaf teeth acute; siliqua linear, 25–70 mm × 3–7 mm; valves thin, dorsally ribbed; rostrum subulate to narrowly conic, 4–11 mm, seedless;44. Petals pale yellow, 13–20 mm × 5–8 mm; siliqua 25–45(–50) mm × 5–7.5 mm;54. Petals yellow, 18–27 mm × 7–13 mm; siliqua 35–70 mm × 3–4.5 mm;65. Petals 14–20 mm × 5–7 mm; fruiting pedicels 10–18 mm, siliqua tetragonous, laterally compressed, 35–45(–50) mm × 6–7.5 mm; seed diameter 2.85–3.2 mm;*B. villosa* subsp. *brevisiliqua*5. Petals 13–18 mm × 6.5–8 mm; fruiting pedicels 16–26 mm, siliqua almost isodiametric, slightly compressed, (20)–25–35(–40) mm × 5–6 mm; seed diameter 2.3–2.5 mm;*B. tardarae*6. Leaves green, glabrous; leaf lamina ovate–lanceolate, acute, deeply incised, margin loosely dentate;*B. rupestris* subsp. *rupestris*6. Leaves with hispid hairs, mainly in upper blade; leaf lamina ovate–elliptical, obtuse, lobed, margin minutely dentate;*B. rupestris* subsp. *hispida*7. Ovary hairy; fruit corpus hairy, subglobose to ellipsoid, smooth, thickened, spongy, 8−18 × 8−11 mm;*B. trichocarpa*7. Ovary glabrous; fruit corpus glabrous, linear–cylindrical, torulose, thin, 25−100 × 36.5 mm;88. Petiole auriculate at the base, up to 15(18) mm;98. Petiole not auriculate, up to 30 mm;109. Pedicels hairy, 14–35 mm; petals yellow; siliqua 60–100 mm × 3–5 mm, not ribbed dorsally; rostrum 10–20 mm; fruiting pedicels 25–35 mm;*B. incana* subsp. *incana*9. Pedicels glabrous, 7–12 mm; petals white; siliqua 30–65 mm × 2.5–3 mm, ribbed dorsally; rostrum 4–10 mm; fruiting pedicels 14–20 mm;*B. incana* subsp. *raimondoi*10. Leaf lamina lyrate, margin crispate–denticulate; petiole winged; siliqua 30–45 mm × 5–6.5 mm;*B. villosa* subsp. *drepanensis*10. Leaf lamina lobed, margin dentate, petiole unwinged; siliqua 3–5 mm wide;1111. Leaf lamina minutely dentate; siliqua laterally compressed, 25–35 × 4–5 mm; *B. villosa* subsp. *tineoi*11. Leaf lamina broadly dentate; siliqua not laterally compressed, 30–75 mm × 3–5 mm;1212. Leaf margin crenate–dentate; sepals 12–15 mm; petals 24–26mm; siliqua 30–60 mm × 4–4.5 mm, dorsally ribbed;*B. villosa* subsp. *villosa*12. Leaf margin irregularly dentate; sepals 8–11 mm; petal 16–22 mm; siliqua 45–75 mm × (3–)3.5–3.8 mm, dorsally not ribbed.*B. villosa* subsp. *bivonana*

## 3. Discussion

The finding of the new species supports Sicily as one of the main centers of diversification of *Brassica* sect. *Brassica* [11,25].

The Tardara population was known (as “*Brassica rupestris*”) since the first work of Snogerup et al. [4]; in fact, the authors of that work talked about an isolated population of *B. rupestris* “10 km N Sciacca”. After that, several authors referred to that population as *B. rupestris* [15,31,32]. As has already happened for other populations in Sicily (e.g., *Brassica bivonana*, previously included in *B. villosa*, and *B. villosa* subsp. *brevisiliqua*, previously considered under *B. rupestris*), our deeper investigations revealed that the Tardara population is clearly different from the other taxa occurring in Sicily and, according to a species concept currently used in this group [10,11], we here describe the new taxon as a new species.

The differences between *Brassica tardarae* and other morphologically similar taxa, namely *B. rupestris* (subsp. *rupestris* and subsp. *hispida*) and *B. villosa* subsp. *brevisiliqua,* are shown above (Table 1). The new species is also well separated from the mentioned similar taxa geographically; the nearest population of a species of *Brassica* sect. *Brassica* (a few km South-East) belongs to *B. villosa* subsp. *bivonana*. From this point of view, the species is isolated and limited to this special site.

The Tardara Gorges deserve some additional remarks. Its cliffs host important and rich plant communities, where our new species is not the only strictly endemic taxon. *Centaurea saccensis* Raimondo et al. [32] is, in fact, another endemite restricted to this site. Interestingly, it seems to be phylogenetically very isolated (and probably older) when compared to the other species of the *Centaurea cineraria* group occurring in Sicily and surrounding areas [34]. If we look at the geology of the site, this part of Sicily was recently considered as part of the foreland, somewhere similar to the south-eastern Sicilian Hyblean area but with a different history [35,36] (Figure 3). Even though other geologists disagree on this view [37,38], there seems to be general agreement that some restricted sites could have (at least partially) emerged in this area over the last few millions years, after the deposition of the “Trubi” pelagic carbonates [39]. This is just a first remark, and further investigations on this topic are ongoing, but it is clear that the paleogeography and phytogeography of Sicily could be revised if this area is confirmed to be crucial for the geological and biological evolution of this part of the Mediterranean; in fact, it could be one of the oldest areas to have emerged during the complex history of Sicily in the last few million years. In other cases, the geology suggested possible solutions to explain the current genetic structure of some species (e.g. *Ambrosinia bassii*, see [40,41]), or was fundamental in the definition of the borders of phytogeographical units [42], but in this case, the importance of this part of Sicily “risks” being greater.

Finally, the Tardara Gorges are a (proposed) geosite but are not protected due to their biological heritage. They are, in fact, not included in any of the parks or reserves or Natura2000 sites occurring in that area. It is clear that its inclusion as a protected area is fundamental to guarantee its peculiar flora and vegetation, so that the enlargement of the nearby Natura2000 site “Complesso Monte Telegrafo e Rocca Ficuzza” (code ITA040006) will be requested to the competent authorities.

## 4. Materials and Methods

Our work was based on field surveys to study the *Brassica* population of the Tardara Gorges. During the field trips, from October 2018 to May 2020, the morphology, ecology, and phenology of the plants in the wild were observed, and were specimens collected. During additional surveys, other populations of similar *Brassica* taxa (*B. villosa* subsp. *brevisiliqua* and *B. rupestris*) were visited and studied. Qualitative characteristics were observed on all reachable plants of the population during the trips. Quantitative characteristics were measured using 20 plants or items (such as pollen grains, stomata, etc.) collected from at least 5 different plants, with calculations of the mean ± standard deviation. Morphology was observed using a stereomicroscope (LeicaMZ9.5, maximum magnification of 60×). Analysis of relevant literature and revision of herbarium specimens were also performed; in detail, specimens preserved in the Herbarium Mediterraneum (PAL) were checked.

For scanning electron microscope (SEM) investigations, air-dried material (pollen grains or fruits) remained untreated and were put on stubs, coated with gold/palladium, and examined using an Oxford Leo 440 SEM.

## Figures and Tables

**Figure 1 plants-09-00947-f001:**
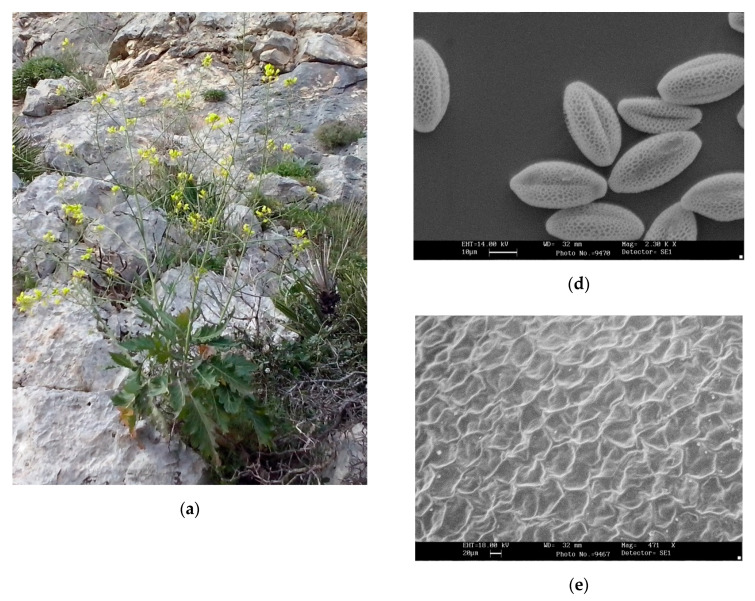
Habitus and morphology of *Brassica tardarae*: (**a**) a plant in its habitat; (**b**) detail of the flower; (**c**) SEM image of a bulbose hair on the blade of the leaf; (**d**) SEM image of pollen grains; (**e**) SEM image of seed surface; (**f**) comparison between siliquae of *Brassica villosa* subsp. *brevisiliqua* (on the left) and *B. tardarae* (on the right); (**g**) seeds of *B. tardarae*.

**Figure 2 plants-09-00947-f002:**
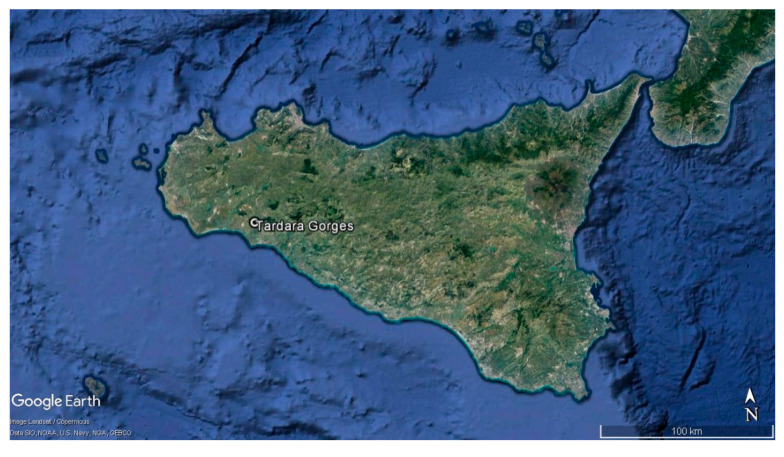
Location of the Tardara Gorges in Sicily.

**Figure 3 plants-09-00947-f003:**
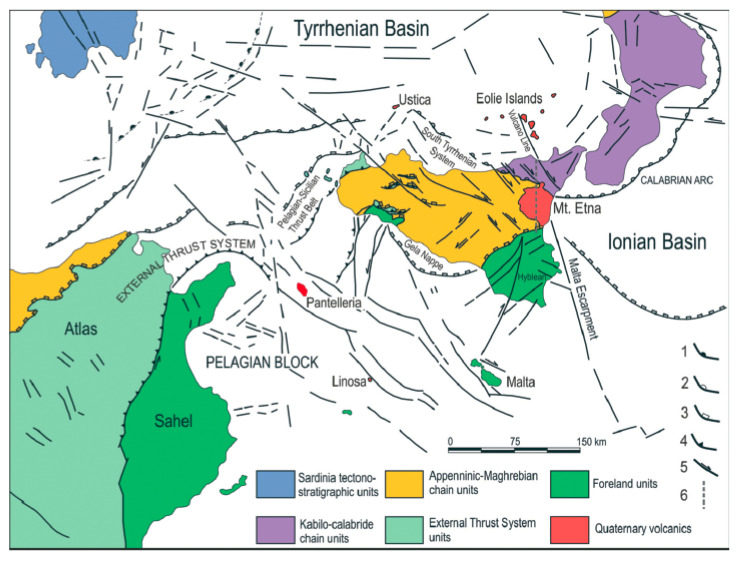
Geological structural map of central Mediterranean Sea (modified from [34], with modification to the color of the island of Pantelleria). Regarding the aim of the present contribution, we note the “Foreland Units” in the area around Sciacca (in western Sicily).

**Table 1 plants-09-00947-t001:** Main features of the studied Tardara Gorges *Brassica* population compared with those of most similar taxa. Main diagnostic characters are in bold. Mean values ± S.D. (where available).

	*Tardara Gorges population*	*Brassica rupestris* subsp. *rupestris*	*Brassica rupestris* subsp. *hispida*	*Brassica villosa* subsp. *brevisiliqua*
Stem indumentum	Glabrous	Glabrous	Glabrous	Glabrous
**Seedling leaves**	Hispid with bulbose hairs	Glabrous with rare bulbose hairs	Hispid with bulbose hairs	Glabrous with rare bulbose hairs
**Adult leaves**	Glabrous, sometimes with rare bulbose hairs in the sterile and in the young shoots, especially on the upper page	Glabrous or with rare bulbose hairs	Hispid with bulbose hairs mainly on the upper epidermis	Glabrous (shiny and thick)
	Basal leaves: ovate–elliptic, sublyrate; petiole in upper part with 1−2 pairs of small lobes; the apical lobe is acute with deep teeth/incisions in the margin	Basal leaves ovate, lyrate, with margin more or less deeply toothed	Basal leaves ovate–elliptic, sublyrate, irregularly toothed	Basal leaves lobate–sublyrate; apical lobe is roundish with margin irregularly crenate
**Petal (mm)**	15.7 ± 1.57 × 7.15 ± 0.41	21.02 ± 1.99 × 9.84 ± 1.35	20.54 ± 2.05 × 9.71 ± 1.32	16,71± 1.02 × 6.68 ± 0.32
**Siliqua**	Short, almost isodiametric, slightly compressed	Long, almost isodiametric, slightly compressed	Long, almost isodiametric, slightly compressed	Intermediate, tetragonous, laterally compressed
**Ripening**	Mid–end of May	Late May–early June	Mid-June	Early May
**Pedicel of siliqua (mm)**	19.92 ± 2.46	13.52 ± 0.57	13.2 ± 2.1	14.02 ± 2.51
**Siliqua length** **(mm)**	30.19 ± 4.56	61.87 ± 3.34	54.1 ± 4.02	40.96 ± 5.04
**Siliqua width (mm)**	5.52 ± 0,42	3.7 ± 0.43	3.3 ± 0.6	6. 89 ± 0.73
Siliqua beak (mm)	4.71 ± 0.41	7.5 ± 2.16	3.8 ± 0.8	5.38 ± 0.65
Seed in the beak	Absent	Absent	1 (Rarely)	Absent
**Ratio** **siliqua/pedicel**	1.53 ± 0.23	4.58 ± 0.26	4.09 ± 0.15	2.99 ± 0.57
Valve, dorsal rib	Present	Slight	Slight	Present
**Inflorescence**	Very elongated inflorescence and infructescence (overall plant up to 1.5–1.8 m tall)	Very elongated inflorescence and infructescence (overall plant up to 1.5 m tall)	Very elongated inflorescence and infructescence (overall plant up to 1.5 m tall)	Contracted (not elongated) inflorescence and infructescence (overall plant up to 1.0–1.2 m tall)
**Taste/flavor**	Bitterish	Pungent/acrid	Bitterish	Pungent/acrid
**Seed diameter (mm)**	2.51 ± 0.11	2.68 ± 0.32	2.5 ± 0.08	3.01 ± 0.10
**Seed color**	Brown–blackish with small light streaks at the hilum	Brown–blackish	Brown–reddish	Brown–blackish
Seed surface	Alveolate–trabeculate (in relief in some parts of the seed)	Generally alveolate–trabeculate	Alveolate–trabeculate (in relief in some parts of the seed)	Alveolate regular
Hilum	A little prominent	Prominent	Prominent	Not prominent
**Pollen length** **× width (µm)**	34.94 ± 1.66 × 18.13 ± 0.72	29.58 ± 0.82 × 16.36 ± 0.37	29.44 ± 1.91 × 16.82 ± 0.75	30.18 ± 0.71 × 16.82 ± 0.74
Pollen ratio (polar/equatorial)	1.93 ± 0.15	1.81 ± 0.07	1.75 ± 0.14	1.80 ± 0.08
Pollen feature	Tricolpate, reticulate, elliptical, tapered at the poles (sometimes with prominence)	Tricolpate, reticulate, elliptical	Tricolpate, reticulate, elliptical, slightly tapered at the poles	Tricolpate, reticulate, elliptical
Stomata on upper blade (n/mm^2^)	297.7 ± 20.6	295.5 ± 10.9	296.4 ± 9.95	242.7 ± 25.8
**Stomata on lower blade (n/mm^2^)**	693.96 ± 53.05	307.0 ± 18.3	325 ± 16.3	553.6 ± 21.48
**Stomata length (µm)**	8.3 ± 1.12	8.2 ± 1.14	8.5 ± 0.98	12.3 ± 1.08

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
