# Peer review of "Brassica tardarae (Brassicaceae), a New Species from a Noteworthy Biotope of South-Western Sicily (Italy)"

_plants, 2020, doi:10.3390/plants9080947_

Round 1

Reviewer 1 Report

Dear Authors,

congratulations for the manuscript, my suggestions are:

The work done is interesting but generates a bit of confusion on the real taxa identified and if this must be considered at the subspecies or species level. No data relating to the descent of the natural population described is reported in order to report the stability of the characteristics described over time. We need much more information on its real interconnection with other cultivated B. oleracea taxa (Hammer, 2013 ...)

Lines 13, 16, 39, 57 ... is to clarify if this taxa represent a species or a subspecies

L28 X = 9 ??? Is better to use n = 9

L42 see the articles Branca et al., 2013 and Maggioni et al.,

L44 12 subspecies taxa ??

Table 1 ... which features are utilized ... could be better use the IPGRI descriptors on Brassicas in order to appear the related data which the several manuscripts published during the last seven years on yje Sicilian wild Brassicas. Is not clear from where this data come from as this details is not reported in the Materials and methods ... Climatic data of the locations considered for the characterization of the morphological data and of their variability (no data are reported about). The characters described, but not scored, are affected by the genetic profile or by the environmental effects? Are this description done on the plants of B. tardarae are done in the same location of the other species / subspecies or in the different locations where the different taxa are widespread?

L104-120… it is possible to confirm these morphological data in the progeny of this population grown in controlled conditions? Are these differential traits among the other populations of B. rupestris and B. villosa in only one location in order to eliminate the environmental effect affects the bio-morphology of the plant (phenotyping) and so taking in consideration only the genetic variability of this population and of the other ones?

L123-126 .. These are the characteristics of some plant organs .. but what about he variability, statistical analysis, standard error, methodology for the analysis, etc. What about the methodology for sampling the organs of the population?

What about the biology and the biological trait of this taxa ???

The discussion do not take into consideration the relevant number of recent articles related to the Brassica oleracea complex species (n = 9) in Sicily and about their bio-morphologial, biochemical and genetic traits. Please consider and cite them. What about the genetic flux occurs in Sicily between the growing and the wild taxa (B. oleracea complex species (n = 9))? This flux could generate new types and forms naturally widespread are new subspecies populations? And in the agrosystems they are cultons? (Branca and Iapichino, 1997).

L240..what about he biology of this taxa?

L245 each instrument and tool need to reported the company name, the city and the country (see the rules for Authors)

L-245-247 no statistical analysis methodology is reported

References ... to be cited the relevant number of articles related to the Brassica oleracea complex species (n = 9) and on their characterization ... a gap cover at least the last seven years ...

Best regards 

Author Response

Dear Authors,

congratulations for the manuscript, my suggestions are:

We thank the reviewer for his/her valuable work on our manuscript. We realize that English is not his/her mother tongue, so we hope we understood all he/she wrote.

The work done is interesting but generates a bit of confusion on the real taxa identified and if this must be considered at the subspecies or species level.

Response: We don’t understand where the confusion is: we explicitly wrote (line 203) that “according to a species concept currently used in this group [10, 11], we described the new taxon as a new species”. In the last years, new taxa in this group (characterized by their morphology, ecology and distribution) have been described as species (Brullo et al. 2013, Sciandrello et al. 2013), and taxa previously considered subspecies have been suggested as species (Tatout et al. 1999).

The conservative solution of attributing the infra-specific rank to these kinds of taxa could hide a recent speciation process useful for other biology facets. Indeed, the long-standing debate on the subspecies usefulness in living world, that fosters a trend to eliminate the trinomial designation in species names, could lead to disregard such taxa. Therefore, we decided to assign the species status.

“As classificatory units, subspecies are not useful in comparative studies, because subspecies are groups of populations defined by hypothetical biological relations or geographical distributions, rather than by homology, that is, shared derived characteristics” (Ebach & Williams, 2009 in Nature 457(7231):785).

No data relating to the descent of the natural population described is reported in order to report the stability of the characteristics described over time. We need much more information on its real interconnection with other cultivated B. oleracea taxa (Hammer, 2013 ...)

Response: The stability of the characters has been verified observing plants in the wild in the last 5 years. We think that the observation of cultivated plants is another work, that is possible to do in addition to (and after) the description of the taxon. In the same way, of course we need more information about relationships with cultivated B. oleracea taxa, but this is another work.

Lines 13, 16, 39, 57 ... is to clarify if this taxa represent a species or a subspecies

Response: We think we are clear in saying that this taxon represents a species, and it is compared with similar taxa.

L28 X = 9 ??? Is better to use n = 9

Response: We preferred to indicate the “x” (base) number, as it is reported in literature (e.g. Quiros et al., 1988, Journal of Heredity 79: 351–358; Snogerup et al. 1990; Maggioni & Alessandrini 2019, Italian Botanist 7: 1–16).

L42 see the articles Branca et al., 2013 and Maggioni et al.,

Response: We made an ad hoc search on Scopus and other bibliographic databases, and we could add to our reference list a single paper (Maggioni et al. 2014): papers of those authors in fact focussed mainly on cultivars and agronomic aspects, and only incidentally on wild taxa and taxonomy, that is the focus of our work.

L44 12 subspecies taxa ??

Response: We used the more generic and inclusive “taxa” because in our list there are both species (e.g. Brassica macrocarpa, B. insularis) and subspecies, so “12 subspecies” should be incorrect

Table 1 ... which features are utilized ... could be better use the IPGRI descriptors on Brassicas in order to appear the related data which the several manuscripts published during the last seven years on yje Sicilian wild Brassicas. Is not clear from where this data come from as this details is not reported in the Materials and methods ... Climatic data of the locations considered for the characterization of the morphological data and of their variability (no data are reported about). The characters described, but not scored, are affected by the genetic profile or by the environmental effects? Are this description done on the plants of B. tardarae are done in the same location of the other species / subspecies or in the different locations where the different taxa are widespread?

Response: As we wrote, descriptions were made on wild plants in their habitat, or on herbarium specimens collected from wild populations. This is the rule for taxonomic papers (see for example Mazzola & Raimondo 1988, Giotta et al. 2002, Brullo et al. 2013, Sciandrello et al. 2013). About IBPGR/IPGRI descriptors, they are conceived for general use and especially (not only) for cultivated morphotypes; we used descriptors (macro- and micro- morphological, and phenological characters) used in recent taxonomic papers and used to discriminate species of this group, so that in this way we are able to compare our data and our new species with previously published data and species.

However, we added more information about the source of our data in Materials & Methods, but not on climatic data that are already reported in other paper that we cite.

L104-120… it is possible to confirm these morphological data in the progeny of this population grown in controlled conditions? Are these differential traits among the other populations of B. rupestris and B. villosa in only one location in order to eliminate the environmental effect affects the bio-morphology of the plant (phenotyping) and so taking in consideration only the genetic variability of this population and of the other ones?

Response: As we said above, we think that the observation of cultivated plants is another work, that is possible to do in addition to (and after) the description of the taxon.

L123-126 .. These are the characteristics of some plant organs ..but what about he variability, statistical analysis, standard error, methodology for the analysis, etc. What about the methodology for sampling the organs of the population?

Response: Thanks for the suggestion. We added something more about these aspects in Materials & Methods.

What about the biology and the biological trait of this taxa ???

Response: We don’t know what the reviewer means for “biological traits”: we described biological traits such as phenology and made reference to other papers describing climate and phytosociology of the site. We added a line about the cliff community where the new species lives.

The discussion do not take into consideration the relevant number of recent articles related to the Brassica oleracea complex species (n = 9) in Sicily and about their bio-morphologial, biochemical and genetic traits. Please consider and cite them. What about the genetic flux occurs in Sicily between the growing and the wild taxa (B. oleracea complex species (n = 9))? This flux could generate new types and forms naturally widespread are new subspecies populations? And in the agrosystems they are cultons? (Branca and Iapichino, 1997).

Response: As we said above, papers of those authors focussed mainly on cultivars and agronomic aspects, and only incidentally on wild taxa and taxonomy, that is the focus of our work. We think that this kind of works can be done after, and not in a paper describing a new species. A possible work could investigate the genetic flow between wild and cultivated populations (as previously made by others, e.g. Maggioni et al. 2014). Here we describe a new species, i.e. a wild population of morphologically well characterized plants.

L240..what about he biology of this taxa?

Response: (“This taxon” – lapsus) See above

L245 each instrument and tool need to reported the company name, the city and the country (see the rules for Authors)

Response: We don’t see such rule in the Plants website, however the only mentioned instrument (SEM) is well described. We added the description of the stereomicroscope.

L-245-247 no statistical analysis methodology is reported

Response: Thanks for the suggestion. We added something more about these aspects in Materials & Methods.

References ... to be cited the relevant number of articles related to the Brassica oleracea complex species (n = 9) and on their characterization ... a gap cover at least the last seven years ...

Response: As we said above, we made an ad hoc bibliographic search, and we could add to our reference list a single paper (Maggioni et al. 2014): papers of those authors in fact focussed mainly on cultivars and agronomic aspects, and only incidentally on wild taxa and taxonomy, that is the focus of our work. This is not a review, and we cited only the papers strictly connected to our work.

Best regards

Reviewer 2 Report

The manuscript I evaluate is very well prepared. The authors describe a new species of the genus Brassica, which they observed during their field trips in  Tardara Gorges.

It contains all the necessary information (morphology, recognition keys, photography, etc.) needed to recognize a new species. I think that interest in this article will not be big, mainly due to the locality of this new species. Nevertheless, I believe that it should be published after incorporating some minor editing amendments.

1. L23-28 and L239-247 Personally, I read such short paragraphs badly.

2. Delete double spaces: e.g. L35, L224

3. L40/L44 Literature order.... ?

4. Be consistent and use the abbreviation: B. instead of Brassica in the rest of the article. L50/L52

5. I suggest removing Figure 2 because it adds nothing.

6. Why don't you introdiuce name of your new species in the manuscript title? I suggest you change the title.

Author Response

Brassica tardarae - Reply to Reviewer 2

Our responses in green

The manuscript I evaluate is very well prepared. The authors describe a new species of the genus Brassica, which they observed during their field trips in  Tardara Gorges.

We thank the reviewer for his/her valuable work on our manuscript and for the positive evaluation.

It contains all the necessary information (morphology, recognition keys, photography, etc.) needed to recognize a new species. I think that interest in this article will not be big, mainly due to the locality of this new species. Nevertheless, I believe that it should be published after incorporating some minor editing amendments.

Response: We think that, although a strict endemic, the new species will have a general interest: because of the importance of the group (section) as crop wild relatives, because of the importance of Sicily as centre of differentiation of this group, because of the biogeographical importance of this new biotope…

  1. L23-28 and L239-247 Personally, I read such short paragraphs badly.

Response: OK, we changed them.

  1. Delete double spaces: e.g. L35, L224

Response: OK, we deleted them.

  1. L40/L44 Literature order.... ?

Response: OK, we changed the order.

  1. Be consistent and use the abbreviation: B. instead of Brassica in the rest of the article. L50/L52

Response: We checked all: we used Brassica only on the first occurrence within a paragraph, then the abbreviated form “B.”.

  1. I suggest removing Figure 2 because it adds nothing.

Response: We think that Figure 2 is important to localize the population, making more clear the following Figure 3.

  1. Why don't you introdiuce name of your new species in the manuscript title? I suggest you change the title.

Response: We accepted the suggestion of the reviewer, changing the title.

Round 2

Reviewer 1 Report

Dear Authors,

the answers are satisfactory, congratulations for the work done.

Best regards